# *In silico* assessment of 18S rDNA metabarcoding markers for the characterization of nematode communities

**Gentile Francesco Ficetola**[1,2], **Alessia Guerrieri**[3], **Isabel Cantera**[1]\*, **Aurelie Bonin**[3]

**1** Department of Environmental Science and Policy, University of Milan, Milan, Italy, **2** LECA, Laboratoire d'Ecologie Alpine, Univ. Grenoble Alpes, CNRS, Univ. Savoie Mont Blanc, Grenoble, France, **3** Argaly, Bâtiment Cleanspace, Sainte-Hélène-du-Lac, France

\* isa_cantera@hotmail.com

**Data Availability Statement:** All the analyses are based on two databases already available online: 18S-NemaBase and GenBank database: -The reference database used is available as supplementary appendix of Gattoni et al., 2023.

## Abstract

Nematodes are keystone actors of soil, freshwater and marine ecosystems, but the complexity of morphological identification has limited broad-scale monitoring of nematode biodiversity. DNA metabarcoding is increasingly used to assess nematode diversity but requires universal primers with high taxonomic coverage and high taxonomic resolution. Several primers have been proposed for the metabarcoding of nematode diversity, many of which target the 18S rRNA gene. *In silico* analyses have a great potential to assess key parameters of primers, including taxonomic coverage, resolution and specificity. Based on a recently-available reference database, we tested *in silico* the performance of fourteen commonly used and one newly optimized primer for nematode metabarcoding. Most primers showed very good coverage, amplifying most of the sequences in the reference database, while four markers showed limited coverage. All primers showed good taxonomic resolution. Resolution was particularly good if the aim was the identification of higher-level taxa, such as genera or families. Overall, species-level resolution was higher for primers amplifying long fragments. None of the primers was highly specific for nematodes as, despite some variation, they all amplified a large number of other eukaryotes. Differences in performance across primers highlight the complexity of the choice of markers appropriate for the metabarcoding of nematodes, which depends on a trade-off between taxonomic resolution and the length of amplified fragments. Our *in silico* analyses provide new insights for the identification of the most appropriate primers, depending on the study goals and the origin of DNA samples. This represents an essential step to design and optimize metabarcoding studies assessing nematode diversity.

## Introduction

Nematodes are probably the most abundant animals on Earth, and are a crucial component of soil, freshwater and marine ecosystems [1–3]. Despite their importance, the complexity and labor of morphological identification has long limited broad-scale analyses of nematode biodiversity [4, 5]. The biodiversity of nematodes is estimated to be 1–10 million species, but less

Journal of Nematology 55 (https://github.com/
WormsEtAl/18SNemaBase) -The GenBank
database is available at https://ftp.ncbi.nlm.nih.gov/
genbank/ -The script are available on figshare via
the following DOI: 10.6084/m9.figshare.24722199.

**Funding:** G.F.F., A.G., I.C. and A.B. have been
funded by the European Research Council under
the European Community's Horizon 2020
Programme, Grant Agreement no. 772284
(IceCommunities). The funders had no role in
study design, data collection and analysis, decision
to publish, or preparation of the manuscript.

**Competing interests:** The authors have declared
that no competing interests exist.

than 30,000 species have been described using morphology [6]. Recent advances in DNA
metabarcoding have fostered the study of nematode biodiversity from a range of environ-
ments, highlighting their impressive diversity and the multiple key roles they play [2, 3, 7–12].

The identification of appropriate primers is a fundamental step of all DNA metabarcoding
analyses [13, 14]. Several features are extremely important for the selection of primers. First,
primers must have a low number of mismatches with the sequences of the target group [high
taxonomic coverage; 15–18]. Second, primers should amplify highly variable regions to enable
the identification of target taxa at a high taxonomic level [high resolution; 16, 17]. Third, short
amplicons are generally favored, in order to reduce the cost of sequencing. Although recent
advances in sequencing technologies now allow to sequence longer fragments with the same
budget [19], the use of primers amplifying short fragments is relevant when working with envi-
ronmental and ancient DNA, as it is often degraded and consists of short sequences [14, 20–
23]. Finally, primers that only amplify the target taxon are frequently preferred (i.e. primers
with high specificitity), because non-specific amplification can reduce the detection of target
taxa, particularly if they show limited abundance [14, 17, 24, 25].

*In silico* approaches are extremely useful to assess key features of primers, including the
number of mismatches with the target sequences (a key determinant of taxonomic coverage)
and the potential taxonomic resolution [17]. *In silico* tests allow cheap and rapid comparisons
of a very large number of primers and often provide a good estimate of the actual performance
of primers across various taxa, that can later be confirmed by *in vitro* assays on real-world sam-
ples [11, 17, 26–29]. Accurate *in silico* assessments of primers for DNA metabarcoding require
the availability of extensive, high-quality reference databases over which primers can be tested
[17, 28]. The recent publication of a curated 18S rRNA database of nematode sequences
[18S-NemaBase; 10] poses the basis for such assessments.

In this study, we built upon the 18S-NemaBase to compare the performance of 15 primers
proposed for the metabarcoding of nematodes (Table 1). Using *in silico* PCR, we 1) assessed
whether the selected primers are able to amplify a large proportion of nematode taxa (cover-
age), 2) tested the taxonomic resolution of primers and evaluated whether there is a trade-off
between marker length and taxonomic resolution and 3) tested primer specificity, i.e., assessed
whether they only amplify nematodes, or also amplify a broad range of other organisms. Our
results help to evaluate the appropriateness of different primers for different aims, ranging
from the analysis of potentially degraded environmental DNA (eDNA) to whole-organism
community DNA [30].

## Methods

### Primer selection

All 15 primers selected for our analyses target the 18S rDNA region of nematodes (Table 1,
Fig 1). Among these primers, 13 were selected because they have been identified by a previous
review as the primers more commonly used in nematode metabarcoding [10]. One additional
primer, Euka02 [31] is often used for metabarcoding of eukaryotic eDNA and it has been sug-
gested to provide a good estimate of nematode diversity [8, 12, 31–33]. Furthermore, we devel-
oped a new primer pair (Nema02; see Table 1) by optimizing primer pair F548_A / R1912
from ref. [11]. More specifically, we performed *in silico* PCRs on the public sequence database
GenBank v249 with the ecoPCR program [17] to evaluate Nematoda and non-Nematoda vari-
ability at each position of the sequences matching the F548_A and R1912 primers, and in a
10-base interval in 5' and 3' of these sequences. The objective was to fine-tune the primer
sequences to maximize non-Nematoda variability while minimizing Nematoda variability in
primer-matching sequences, especially in 3', in order to increase specificity. Taxonomic

**Table 1. List of primer pairs tested *in silico*, forward and reverse sequences of primers and the expected minimum and maximum lengths (bp) of the amplicons used in the ecoPCR program.**

| PRIMER | Forward sequence | Reverse sequence | min—max amplicon length (bp) | Reference |
|---|---|---|---|---|
| 1391f- EukBr | GTACACACCGCCCGTC | TGATCCTTCTGCAGGTTCACCTAC | 50–250 | [41, 42, 72] |
| 1813F—2646R | CTGCGTGAGAGGTGAAAT | GCTACCTTGTTACGACTTTT | 500–1000 | [44] |
| 18SILVOmidF—18SILVOmidR | CAAGTCTGGTGCCAGCAG | GAGTCTCGCTCGTTATCGG | 500–1000 | [48] |
| 3NDf—1132rmod | GGCAAGTCTGGTGCCAG | TCCGTCAATTYCTTTAAGT | 300–700 | [51] |
| EcoF—EcoR | GGTTAAAAMGYTCGTAGTTG | TGGTGGTGCCCTTCCGTCA | 300–700 | [48] |
| Ek-NSF573—Ek-NSR951 | CGCGGTAATTCCAGCTCCA | TTGGYRAATGCTTTCGC | 200–500 | [73] |
| Euka02 | TTTGTCTGSTTRATTSCG | CACAGACCTGTTATTGC | 30–400 | [31] |
| F_1183 –R_1631 | AATTTGACTCAACACGGG | TACAAAGGGCAGGGACG | 300–600 | [74, 75] |
| FO4—R22 | GCTTGTCTCAAAGATTAAGCC | GCCTGCTGCCTTCCTTGGA | 200–500 | [40] |
| MMSF—MMSR | GGTGCCAGCAGCCGCGGTA | CTTTAAGTTTCAGCTTTGC | 300–700 | [49] |
| Nema02 | AAGTCTGGTGCCAGCAGC | GTTTACGGTYAGAACTAGGG | 325–801 | This study, modified from Kawanobe et al. [11] |
| NemF—18Sr2b | GGGGAAGTATGGTTGCAAA | TACAAAGGGCAGGGACGTAAT | 300–700 | [4, 5] |
| NemFopt—18Sr2bopt | GGGGWAGTATGGTTGCAAA | TGTGTACAAAKGRCAGGGAC | 300–700 | [48] |
| NF1—18Sr2b | GGTGGTGCATGGCCGTTCTTAGTT | TACAAAGGGCAGGGACGTAAT | 200–500 | [4, 5] |
| SSU_F04—SSU_R22 | GCTTGTCTCAAAGATTAAGCC | CCTGCTGCCTTCCTTGGA | 200–500 | [43] |

resolution of the associated marker was evaluated using the ecostaxspecificity program of the OBITools package [34]. Compatibility of annealing temperatures and absence of problematic primer dimers or hairpins were checked using OligoCalc [http://biotools.nubic.northwestern.edu/OligoCalc.html; 35] and the OligoAnalyzer Tool (https://eu.idtdna.com/pages/tools/oligoanalyzer?returnurl=%2Fcalc%2Fanalyzer), respectively.

### *In silico* PCR

The availability of high-quality, curated databases is pivotal to test the performance of metabarcoding primers. We based our analyses on the 18S-NemaBase [10], which represents the

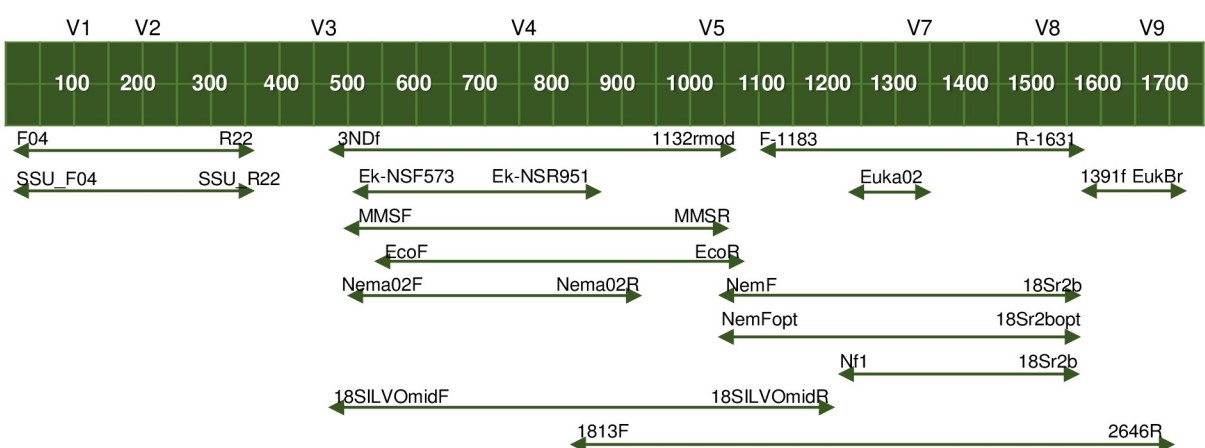

**Fig 1. Location of primers tested for nematode metabarcoding within the 18S rDNA gene.** Primers were aligned with the *Caenorhabditis elegans* 18S rDNA gene (GenBank accession number: AY268117). Redrawn on the basis of ref. [10].

most complete and high-quality available reference database of nematode 18S rRNA. The original 18S-NemaBase includes 5231 sequences identified at family level or better, representing 214 families and 2734 species [10].

The performance of primers was tested using the ecoPCR program [17]. EcoPCR allows *in silico* assessment of amplification of a sequence on the basis of its match with a selected primer pair in a region of a specified length [17]. The sequence is selected from a given reference database. To work, the program requires the reference database in the ecoPCR format. Thus, we converted the 18S-NemaBase from the fasta to the ecoPCR format using the obiconvert command of the OBITools command suite [34]. The original 18S-NemaBase database consisted of 5231 sequences. However, 283 sequences (i.e. 5,4%) were excluded because they showed problems during the conversion from the fasta to the ecoPCR format (268 sequences) or because they were assigned to non-nematode taxa in 18S-NemaBase (15 sequences). Thus, we based our analyses on a total of 4948 sequences.

For *in silico* PCR, we allowed a maximum of 3 mismatches between each primer and the sequences in the database. For each primer pair, the length of amplified fragments was selected based on the literature (Table 1).

Running ecoPCR on the curated 18S-NemaBase allowed us to calculate two measures of primer performance:

- Taxonomic coverage, as the percentage of amplified sequences within the reference database;

- Resolution, i.e. the ability of markers to distinguish between closely-related taxa.

Taxonomic resolution was calculated using the procedure detailed in ref. [36]. First, all the sequences obtained in each ecoPCR were compared among them to produce a list of unique metabarcodes. We then obtained the list of taxa associated to each unique metabarcodes. Taxonomic resolution was tested at three levels: species, genus and family. We assessed if, for each unique metabarcodes and taxonomic level, all the amplified taxa belong to the same taxon. Let's assume, for instance, that multiple species within one genus share the same metabarcode. This particular metabarcode shows a genus-level and a family-level resolution but not a species-level resolution. The average taxonomic resolution of markers was then calculated as the proportion of unique metabarcodes that have a species-level, genus-level and family-level resolution [36].

We used a linear regression to test whether there is a positive relationship between taxonomic resolution and the log-transformed average length of the amplified fragments. This analysis was run at the species-level resolution because all the primers showed excellent resolution at the genus- and family level (see Results).

Furthermore, we assessed the specificity of the tested primers using the whole GenBank database (version 249) instead of the 18S-NemaBase as reference database in the ecoPCR program. This time, we only retained one sequence per species, to avoid biases due to the overrepresentation of model species (e.g. *Caenorhabditis elegans* for nematodes). Then, specificity was measured as the percentage of nematodes amplified over non-target organisms [17]. Primers showing high specificity are in principle preferable because they have a higher probability of detecting target taxa, including rare species [14, 17, 28, 37, 38].

Finally, we used primer sequence logos (weblogos) to further assess the conservation of primers both in nematodes and in non-target taxa [14, 39]. In weblogos, we graphically represented the pattern of primer conservation, by retrieving the match of the forward and reverse primers against GenBank sequences, based on the results of the ecoPCR run over GenBank v249. Logos were built as stacks of symbols (A, C, G, T), with one stack for each position in the

primer sequence. The height of each stack corresponds to the nucleotide conservation at each position, measured in bits and ranging from 0 (same probability for the four nucleotides) to 2 (perfect conservation of the position) [39]. Like specificity analyses (see above), weblogo analyses were run after randomly selecting only one amplicon per species, to avoid bias due to the overrepresentation of model species in GenBank. Weblogos were built using the ggseqlogo R package [39].

## Results

### Taxonomic coverage

The taxonomic coverage (proportion of amplified sequences compared to the 18S-NemaBase database) was highly variable across primers. The majority of primers showed very good to excellent coverage, amplifying >90% of sequences in the reference database, while four primers showed a limited coverage (Table 2, Fig 2). The three primers with the highest coverage were 3NDf-1132rmod, EcoF-EcoR and Euka02, all showing coverage ≥ 97%. These primers amplified fragments with very different lengths, spanning from 100 bp (Euka02) to more than 800 bp (1813F_2646R; Table 2).

### Taxonomic resolution

All primers showed a good resolution if the aim was species-level identification. Even the primer with the lowest species-level resolution (Euka02), associated most (78%) of unique metabarcodes with just one nematode species (Table 2, Fig 3). Eight primers showed a species-level resolution >90%; all of them amplified fragments >350bp. The three primers with the highest species-level resolution (95%) showed generally low coverage (Table 2, Figs 2 and 3).

If the aim was genus-level identification, all primers showed excellent resolution, as they were able to tease apart 95% of genera or more. Resolution was even better if the target was family identification, with all primers showing a resolution ≥97% (Table 2, Fig 3). Overall, we observed a positive relationship between primer taxonomic resolution at the species level, and

**Table 2. Results of *in silico* PCRs testing the taxonomic coverage and the resolution of 15 primer pairs proposed for nematode metabarcoding.** Metabarcode length refers to the length of the amplified fragment.

| Primer | Taxonomic | Metabarcode length (bp) | | Taxonomic resolution | | |
|---|---|---|---|---|---|---|
| | coverage | mean | Range | Species | Genus | Family |
| 1391f-EukBr | 0.10 | 129.4 | 87–138 | 0.87 | 0.96 | 0.97 |
| 1813F-2646R | 0.36 | 844.9 | 760–987 | 0.95 | 1.00 | 1.00 |
| 18SILVOmidF-18SILVOmid | 0.96 | 742.3 | 664–905 | 0.92 | 0.99 | 1.00 |
| 3NDf-1132rmod | 0.97 | 562.3 | 484–697 | 0.91 | 0.99 | 1.00 |
| EcoF-EcoR | 0.98 | 508.3 | 430–671 | 0.91 | 0.99 | 1.00 |
| Ek_NSF573-Ek_NSR95 | 0.96 | 343.6 | 274–446 | 0.89 | 0.98 | 0.99 |
| Euka02 | 0.97 | 99.2 | 50–330 | 0.78 | 0.95 | 0.99 |
| F_1183-R_1631 | 0.92 | 414.4 | 331–565 | 0.86 | 0.98 | 0.99 |
| FO4-R22 | 0.14 | 375.5 | 281–413 | 0.95 | 1.00 | 1.00 |
| MMSF-MMSR | 0.97 | 540.3 | 462–675 | 0.91 | 0.99 | 1.00 |
| Nema02 | 0.94 | 435.6 | 357–599 | 0.90 | 0.98 | 1.00 |
| NemF-18Sr2b | 0.91 | 484.4 | 401–635 | 0.87 | 0.98 | 0.99 |
| NemFopt-18Sr2bopt | 0.91 | 489.4 | 406–640 | 0.87 | 0.98 | 0.99 |
| NF1-18Sr2b | 0.91 | 320.4 | 237–471 | 0.85 | 0.98 | 0.99 |
| SSU_F04-SSU_R22 | 0.14 | 375.5 | 281–413 | 0.95 | 1.00 | 1.00 |

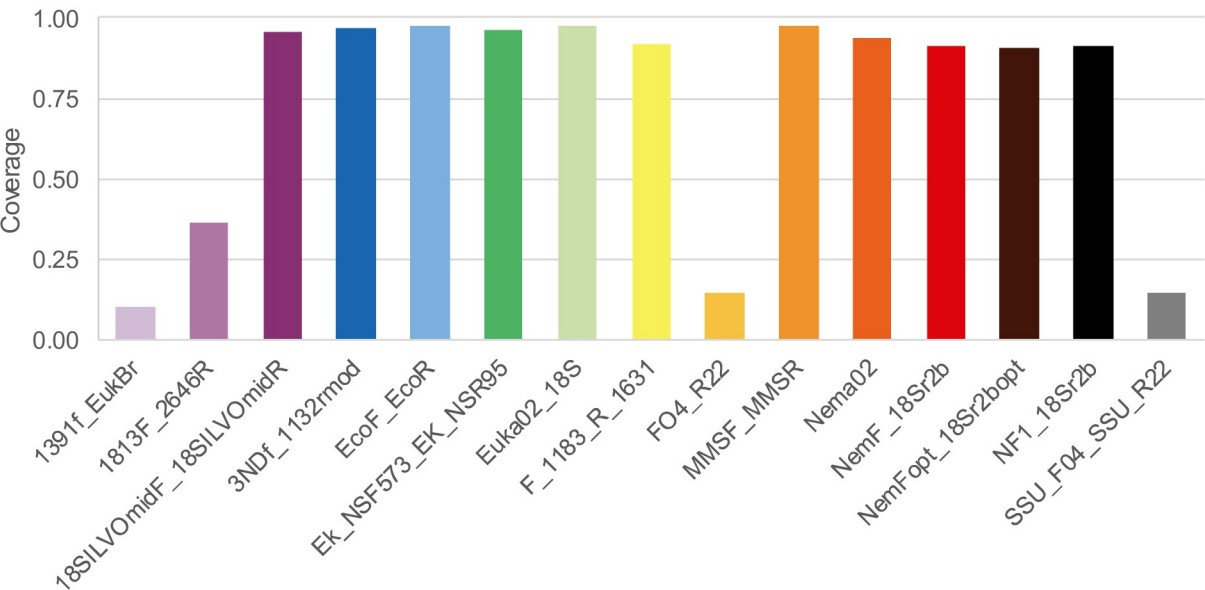

**Fig 2. Taxonomic coverage (proportion of amplified sequences) of the 15 primer pairs tested *in silico* on the 18S-NemaBase.**

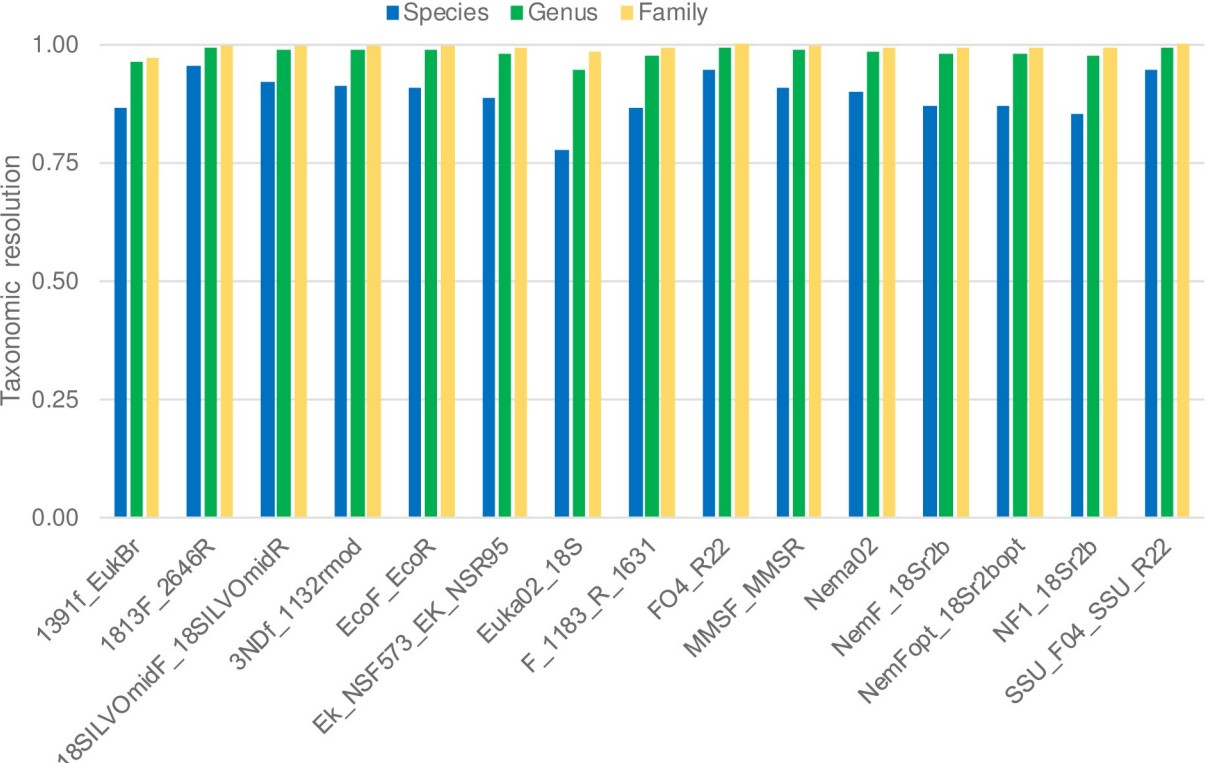

**Fig 3. Species-, genus- and family- level taxonomic resolution of the 15 primer pairs tested *in silico* on the 18S-NemaBase.**

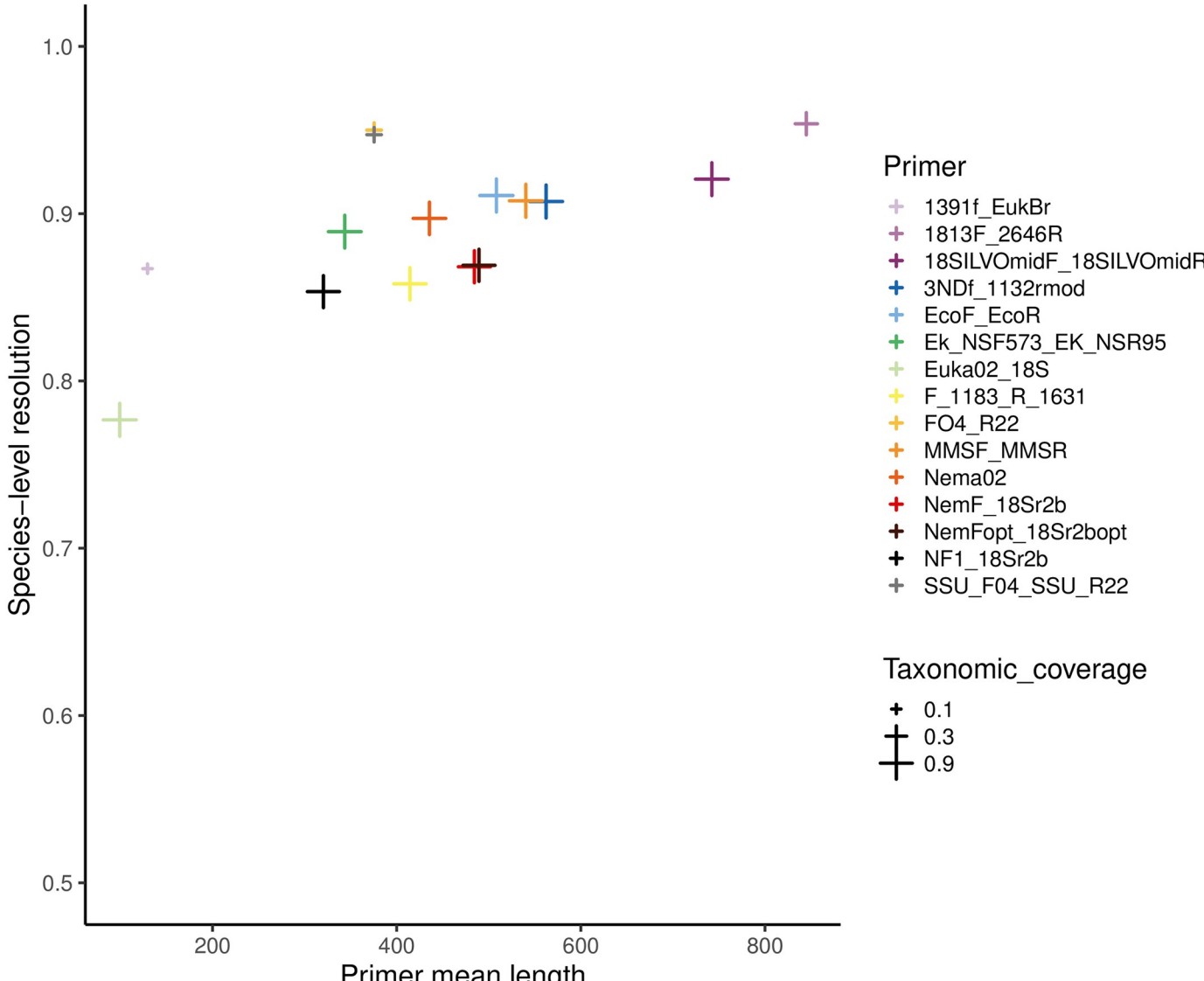

**Fig 4. Relationship between mean length (bp) of amplified fragments (excluding primers) and species-level resolution of the 15 primer pairs.** The size of symbols is proportional to primer coverage.

the length of the amplified metabarcodes (linear regression: $F_{1,13} = 13.6$, $P = 0.003$; $R^2 = 0.51$; Fig 4), suggesting that the increase in the metabarcodes length facilitate taxonomic resolution at the species-level.

## Specificity

When tested on the whole GenBank, primers showed specificity values that ranged between 1% or less (FO4-R22, SSU_F04-SSU_R22) and ~7% (1813F-2646R, Nema02; Fig 5a). These values indicate that all the markers amplify a very large number of non-nematode sequences. Nevertheless, some of the primers with high specificity only amplified a limited number of nematode taxa (Figs 1 and 5b). The primers with the best compromise between specificity and

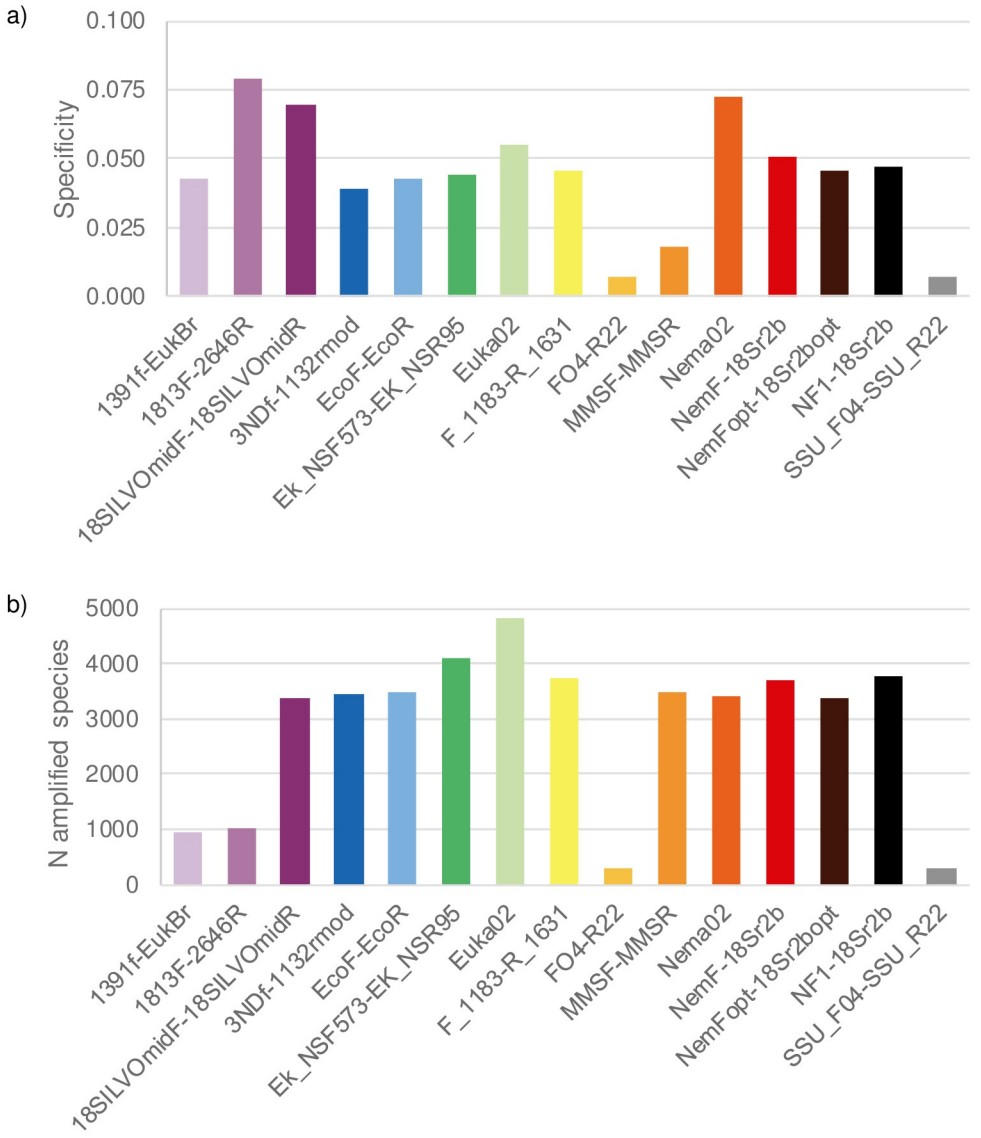

**Fig 5. Specificity of the 15 primer pairs tested *in silico* on the whole Genbank database.** a) Specificity was measured as the N of nematode sequences / total N of amplified sequences. b) N of nematode species amplified by the different primer pairs, when the *in silico* PCR is run on the whole GenBank.

coverage included 18SILVOmidF-18SILVOmid, Euka02 and Nema02 (Figs 3–5). For both nematodes and non-nematodes, weblogos showed clear differences in the patterns of variability of bases in the regions matching the tested primers (Figs 6 and 7).

## Discussion

Our analyses showed that primers suggested for the assessment of nematode diversity have heterogeneous performances, particularly in terms of taxonomic coverage and specificity. Conversely, the taxonomic resolution of all primers was generally good, and across-primer variation in resolution was strongly related to the well-known trade-off with metabarcode length (Fig 4). The selection of the most appropriate primer pair for the analyses of nematode

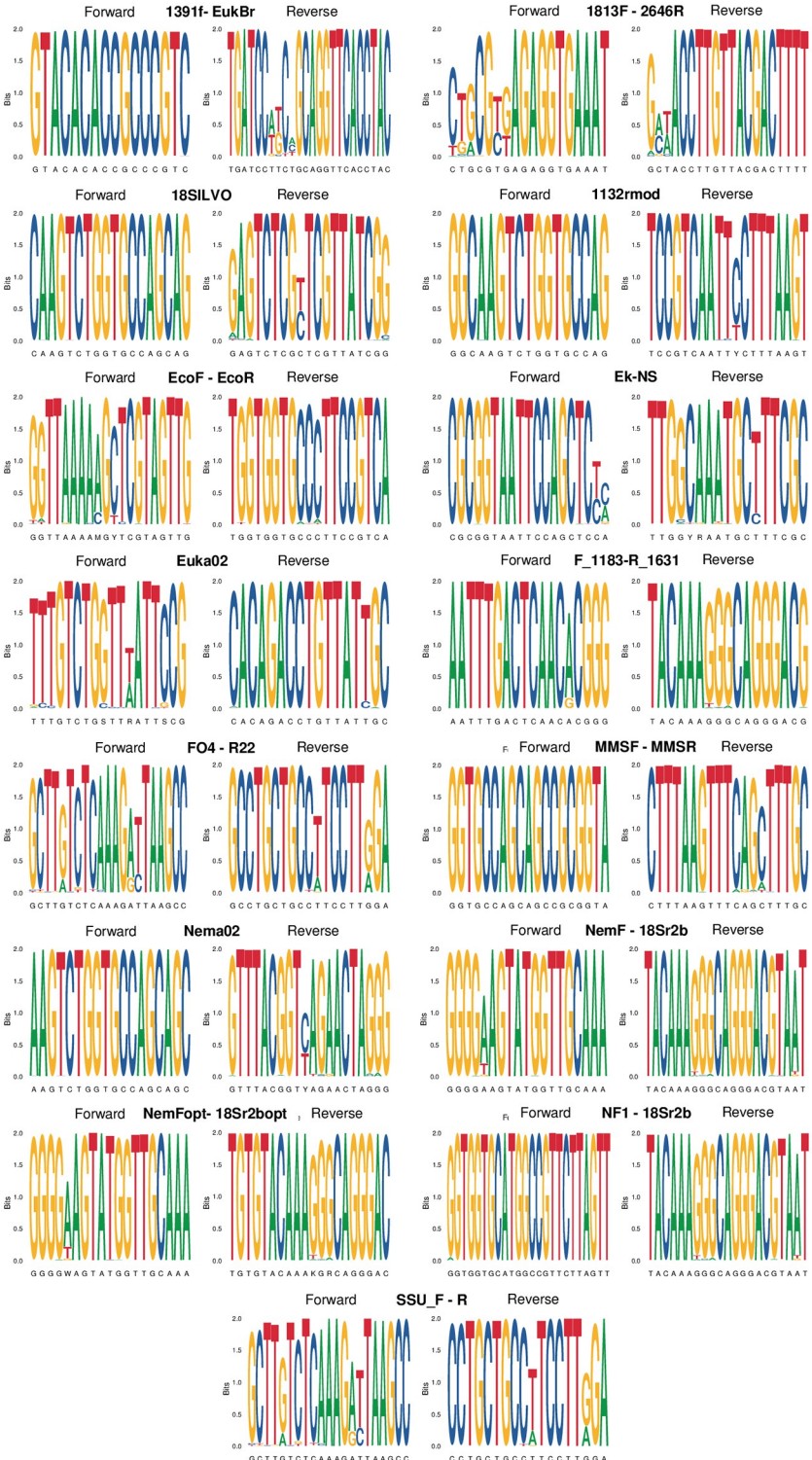

**Fig 6. Weblogos built on primer sequences from nematodes.** The height of stacks corresponds to the nucleotide conservation at that position; the height of symbols within stacks indicates the relative frequency of each nucleotide at that position.

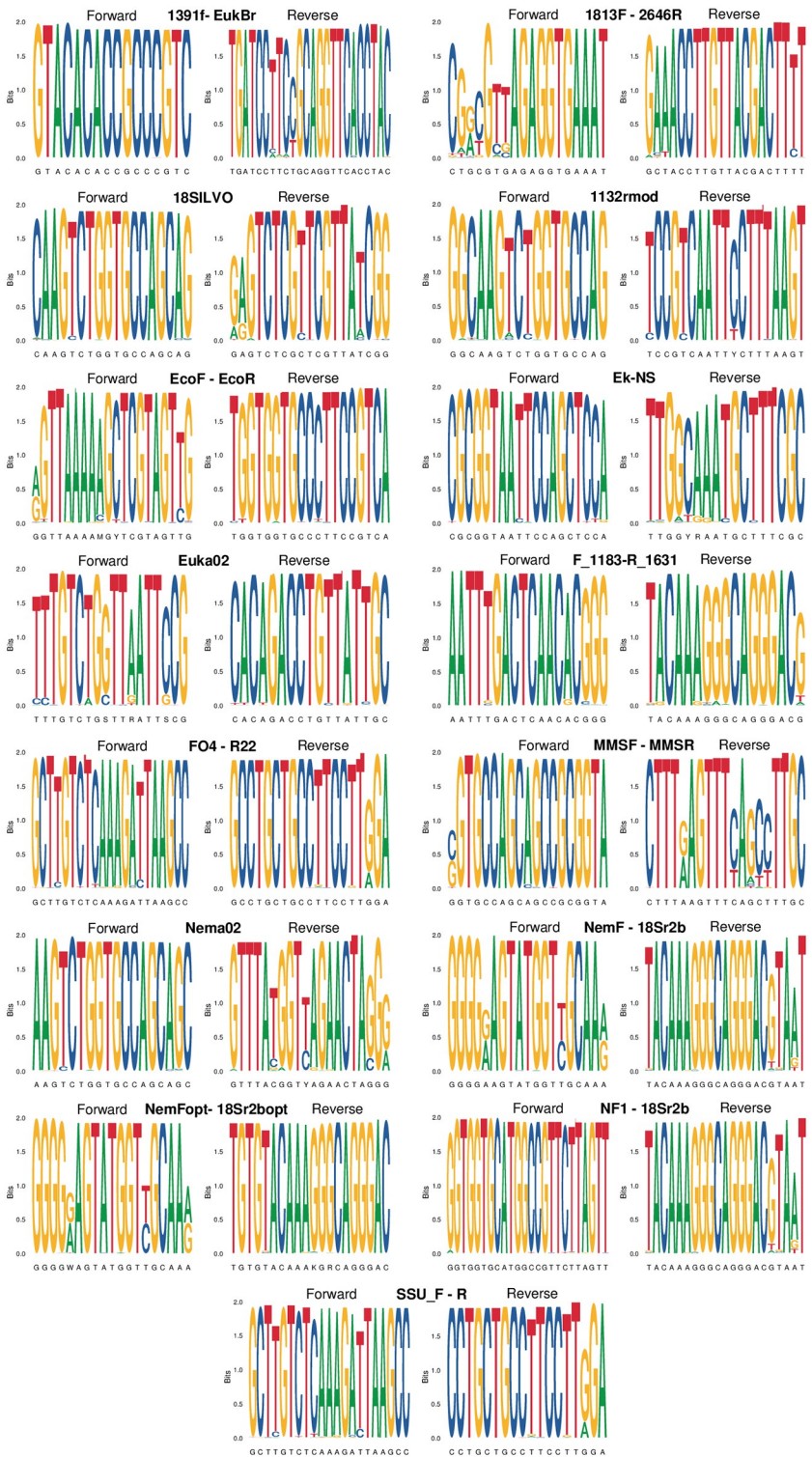

**Fig 7. Weblogos built on primer sequences from non-nematodes.** The height of stacks corresponds to the nucleotide conservation at that position; the height of symbols within stacks indicates the relative frequency of each nucleotide at that position.

biodiversity depends on the balance between multiple factors, including study aims, media from which DNA is extracted, and the adopted sequencing technology.

The taxonomic coverage was highly variable. The majority of primers showed very good to excellent coverage, as they amplified more than 90% (and sometimes >97%) of available sequences, still, some showed limited coverage. The primers with low coverage were either universal primers that amplify a large number of eukaryotes and are not designed specifically for nematodes (FO4—R22, 1391f- EukBr) [40–42], or primers from landmark studies of nematode phylogeny (1813F - 2646R, SSU_R22) [43, 44] that were developed using sequences available at that time. It should be highlighted that our bioinformatic analyses are based on the number and position of mismatches in the priming region. This parameter is particularly relevant in metabarcoding studies, when DNA is amplified by complex mixtures comprising the DNA of many different taxa. In these cases, a large number of mismatches is problematic. Species with less mismatches are amplified preferentially, while species with more mismatches, or with mismatches close to the 3' end, tend to be overlooked [15, 16, 45, 46]. A few mismatches probably are less problematic in phylogenetic studies, where the DNA of just one species is extracted at each time directly from specimens [43, 44].

*In silico* assessments of primer coverage, like the one performed here, have their own limitations, as some taxa may be amplified *in vitro* but not *in silico*, and vice versa. *In silico* analyses focusing on curated databases can also miss key issues, such as non-specific amplification [28]. Nevertheless, several *in vitro* tests have confirmed the appropriateness of primers studied with *in silico* analyses [11, 26–28, 47, 48]. For instance, our results are in agreement with previous studies that used mock communities to compare the performance of MMSF-MMSR, NemF-18Sr2b, NF1_18Sr2b and SSU_F04-SSU_R22, and observed a limited coverage of SSU_F04-SSU_R22 [18, 49]. Similarly, Kenmotsu et al. [50] confirmed that F_1183-R_1631 and NF1_18Sr2b show a similar, very good performance; Geisen et al. [51] confirmed the excellent performance of 3NDf-1132rmod; and Guardiola et al. [31] suggested an excellent coverage for Euka02 [see also ref. 36 for *in vitro* tests confirming the excellent taxonomic coverage of this marker for all the tested invertebrate phyla]. Nevertheless, comparative *in vitro* tests performed on both mock communities and real samples will be extremely important to validate our conclusions on taxonomic coverage, particularly for primers that have received limited testing so far [e.g., Nema02, but see references in 11 for analyses confirming the good performance of related primers].

All primers showed good to excellent resolution on the considered reference database. Even the shortest primer (Euka02) showed a reasonably good ability to discriminate between species (Fig 3) and, for all primers, the frequency of genera sharing the same metabarcode was ≤5% (i.e. genus-level resolution was always 95% or higher). These findings are highly promising for the use of metabarcoding for nematode analyses. However, it is important to acknowledge some caveats. The use of high quality, curated databases is fundamental for all the metabarcoding analyses, being pivotal for assessments of marker performance, and for accurate taxonomic identification [14, 52–55]. Our analyses were run on a large, curated database containing about 5000 sequences from 214 families, 668 genera and 2734 species [10]. Unfortunately, this database only represents the currently described species and genera, but most nematodes inhabiting the Earth still require description [6, 56]. We stress that all measures of taxonomic resolution strongly depend on the available data [57]. For instance, if the reference database only includes one species within a given genus, analyses would return a species-level resolution, despite unanalysed species within that genus may share the same metabarcode [36]. Less optimistic resolution values might be obtained with broader reference databases.

A key issue of the analysed primers is that none of them is specific to nematodes, and they amplify a broad range of eukaryotes (Fig 5). Non-specific amplification can reduce the

detection of rare taxa, and can even increase false positives [28, 37]. This can be particularly problematic when analyses target complex mixtures of DNA(e.g. eDNA extracted from soil) that comprise the DNA of both nematodes and other organisms [58, 59]. For instance, *in vitro* assessments of the EcoF-EcoR primers detected a very large number of non-nematode taxa, suggesting that this marker can be not appropriate for analyses only focusing on nematodes [48]. The low specificity of most primers make them valuable for whole analyses of eukaryote diversity in soil, sediments or aquatic environments [31, 40, 48]. Studies focusing on nematodes should thus assess whether the retrieved data are enough to obtain reliable estimates of species diversity / occurrence. Approaches such as rarefaction curves and analyses of detection probability can allow to assess whether key parameters, such as the number of replicates and sequencing depth, are enough to obtain robust biodiversity estimates, or need to be increased for well-grounded ecological inference [58, 60, 61]. Alternatively, fine-tuning the primer sequences so as to introduce variability at the 3'end of the sequence for non-nematode can greatly help increase the real marker specificity. For example, preliminary analyses of soil samples with the Nema02 marker showed that >50% of MOTUs and >70% of sequences were assigned to nematodes, despite the limited specificity of this marker *in silico*.

Within the primers with good taxonomic coverage, resolution was clearly related to the length of the amplified fragment (Fig 4). This is not unexpected, as longer fragments generally include more variable sites and thus can provide a better resolution [14; see below for further discussion]. Nevertheless, even the shortest fragments allow an excellent genus-level resolution. Many functional analyses of nematodes are performed at the genus level. For instance, Nemaplex (http://nemaplex.ucdavis.edu) is a major database of nematode traits, and provides traits at the genus-level resolution. Ensuring that primers provide robust identification at the genus level is thus extremely important for all the studies focusing on nematode traits and functional diversity.

The better resolution of markers amplifying long fragments is related to a frequent trade-off between taxonomic resolution and fragment length. Long markers generally have a larger number of informative bases, and are therefore expected to show a better capacity of discrimination between closely-related species. For instance, many barcoding studies use a standard marker amplifying a 658-bp long fragment of COI, because it is assumed to provide enough resolution to discriminate between closely-related species [62]. Nevertheless, the study of long markers can be problematic in some conditions. First, DNA extracted from difficult substrates (e.g., environmental DNA from sediments and water) or from museum specimens is often degraded, making the use of long markers challenging [14, 62–65]. Furthermore, some high-throughput sequencing technologies are particularly cost-effective for sequencing short amplicons (e.g. Illumina NovaSeq), while cost-effectiveness decreases and error rate increases if longer fragments are targeted [62]. Therefore, when planning their study, researchers must find the right balance considering study aims, targeted substrate and cost-effectiveness. Our analysis can help to identify the most appropriate primer pair depending on the study focus. Markers amplifying short fragments (<150 bp) can be preferred by studies extracting DNA from water or ancient sediments [66] or if the aim is the analysis of a massive amount of specimens that require highly cost-effective sequencing platforms [62]. Longer fragments (within 450–500 bp) can be appropriate for both metabarcoding of whole-organism community DNA, for intracellular eDNA, and for eDNA extracted from substrates protecting eDNA from degradation (e.g. developed soils with high clay content) [67, 68]. These fragments can be sequenced with platforms such as Illumina MiSeq, which still enable processing a large number of samples at reasonable prices. Primers well suited for this aim include 3NDf-1132rmod, MMSF-MMSR and, if confirmed by *in vitro* tests, Nema02. Among these primers, Nema02 is the one with the highest specificity for nematodes (Fig 5a and 5b). Finally, long fragments

ensure the highest resolution and are particularly suited for whole-organism community DNA [but see 19 for a remarkable example of long-read analysis of DNA extracted from environmental samples]. It has been suggested that, beyond a given threshold, longer markers do not provide a better resolution because of the occurrence of a saturation point [62]. For nematodes, we did not detect such saturation point, as the highest resolution tended to increase with marker length also above 500 bp (Fig 4). The continuing developments of high-throughput sequencing are making long-read metabarcoding increasingly feasible at progressively more affordable prices.

Ongoing development of DNA metabarcoding are opening new avenues to the study of biodiversity, and to the identification of management priorities [12, 69–71]. Nematodes are increasingly recognized as key components of soil communities, and advances in molecular approaches to assess and monitor their biodiversity are extremely important for the growing knowledge on this highly diverse phylum [11, 56]. The selection of most appropriate markers is a fundamental step to maximise the information drawn by each study [61].

## Acknowledgments

The authors acknowledge the support of the APC central fund of the University of Milan.

## Author Contributions

**Conceptualization:** Gentile Francesco Ficetola, Alessia Guerrieri, Isabel Cantera, Aurelie Bonin.

**Data curation:** Alessia Guerrieri, Aurelie Bonin.

**Formal analysis:** Gentile Francesco Ficetola, Alessia Guerrieri.

**Funding acquisition:** Gentile Francesco Ficetola.

**Investigation:** Gentile Francesco Ficetola, Alessia Guerrieri, Aurelie Bonin.

**Methodology:** Alessia Guerrieri.

**Resources:** Aurelie Bonin.

**Software:** Aurelie Bonin.

**Supervision:** Gentile Francesco Ficetola.

**Writing – original draft:** Gentile Francesco Ficetola.

**Writing – review & editing:** Alessia Guerrieri, Isabel Cantera, Aurelie Bonin.

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
