## [Decision Letter · Decision Letter 0]

9 Jan 2024

PONE-D-23-28890In silico assessment of 18S rDNA metabarcoding markers for the characterization of nematode communitiesPLOS ONE

Dear Dr. Ficetola,

Thank you for submitting your manuscript to PLOS ONE. After careful consideration, we feel that it has merit but does not fully meet PLOS ONE’s publication criteria as it currently stands. Therefore, we invite you to submit a revised version of the manuscript that addresses the points raised during the review process.

I got the recommendations and comments from an expert reviewer in the field. Both reviewers agree that the manuscript is technically sound and the data support the conclusions.

However, the lack of an explanation in the Methods, and Results and presentation was suggested by a reviewer, and I shared the comments. Therefore, I can invite you to submit a revised version of the manuscript that addresses the points raised by the reviewers.

We look forward to receiving your revised manuscript.

Kind regards,

Hideyuki Doi

Academic Editor

PLOS ONE

“This study was funded by the European Research Council under the European Community’s Horizon 2020 Programme, Grant Agreement no. 772284 (IceCommunities).”

“G.F.F., A.G., I.C. and A.B. have been funded by the European Research Council under the European Community’s Horizon 2020 Programme, Grant Agreement no. 772284 (IceCommunities).

Additional Editor Comments:

I got the recommendations and comments from an expert reviewer in the field. Both reviewers agree that the manuscript is technically sound and the data support the conclusions.

However, the lack of an explanation in the Methods, and Results and presentation was suggested by a reviewer, and I shared the comments. Therefore, I can invite you to submit a revised version of the manuscript that addresses the points raised by the reviewers.

Reviewers' comments:

Reviewer's Responses to Questions

**Comments to the Author**

1. Is the manuscript technically sound, and do the data support the conclusions?

Reviewer #1: Yes

2. Has the statistical analysis been performed appropriately and rigorously? 

Reviewer #1: Yes

3. Have the authors made all data underlying the findings in their manuscript fully available?

Reviewer #1: Yes

4. Is the manuscript presented in an intelligible fashion and written in standard English?

Reviewer #1: Yes

5. Review Comments to the Author

Reviewer #1: The paper addresses the crucial role of nematodes as keystone actors in various ecosystems and emphasizes the limitations posed by the complexity of morphological identification for monitoring their biodiversity on a broad scale. The adoption of DNA metabarcoding has emerged as a promising approach for assessing nematode biodiversity; however, the success of this method hinges on the utilization of universal primers with high taxonomic coverage and resolution. The authors undertake a comprehensive in-silico analysis of fourteen commonly used and one newly optimized primer for nematode metabarcoding, leveraging a recently available reference database. The evaluation encompasses key parameters such as taxonomic coverage, resolution, and specificity. The findings reveal that while most primers demonstrate excellent coverage, a subset exhibits limited coverage. Notably, all primers exhibit good taxonomic resolution, particularly at higher taxonomic levels like genera or families. The study underscores that species-level resolution tends to be higher for primers amplifying longer fragments. One notable observation is the lack of high specificity for nematodes across all primers, as they amplify a substantial number of other eukaryotes. This insight emphasizes the complexity in choosing markers appropriate for nematode metabarcoding, necessitating a trade-off between taxonomic resolution and amplified fragment length. In conclusion, the in-silico analyses presented in this paper provide valuable insights into selecting optimal primers based on study goals and the origin of DNA samples. This work represents a critical step in the design and optimization of metabarcoding studies aimed at assessing nematode diversity. The introduction effectively contextualizes the importance of nematodes in ecosystems, and the paper overall contributes to advancing our understanding of nematode biodiversity using molecular techniques.

However, I think a major flaw of this study is for comparing the primer specificity. The study mainly showed the primer specificity using specificity value. But I think which base on the primer has mismatching would be more important. Also, I think the trade-off between taxonomic resolution and the length of amplified fragments is a key observation. The paper should provide a more detailed discussion on the practical implications of this trade-off in the context of different research goals and objectives. Recommendations for researchers navigating this trade-off would be beneficial.

Comments on figures

As new Figure 1, It would be better to show a figure for the DNA regions in 18S with the primer regions.

Figure 3: We acknowledge the difficulty in discerning coverage differences in Figure 3. To address this concern, we plan to enhance the visualization by adjusting the color scheme, possibly employing contrasting colors or patterns to highlight distinctions more effectively. This modification aims to provide a clearer representation of the coverage variations, ensuring that readers can easily interpret and compare the data.

Figure 4 and Others: In response to the suggestion of using different colors for the gene regions in Figure 4 and subsequent figures, we find this to be a valuable enhancement.

6. PLOS authors have the option to publish the peer review history of their article (what does this mean?). If published, this will include your full peer review and any attached files.

Reviewer #1: No

---

## [Author Response · Author response to Decision Letter 0]

31 Jan 2024

Dear editors,

We are glad of the positive comments on our manuscript entitled “In silico assessment of 18S rDNA metabarcoding markers for the characterization of nematode communities”. The manuscript has been modified following the reviewer suggestions. Specifically, we updated the figures following the Reviewer indications, we added new figures (weblogos) to show the mismatches in the primer region, and expanded the discussion on the trade-off between marker length and resolution. All the material needed to run analyses is now available online on Figshare.

We hope the manuscript will be now more suitable for publication with PlosOne

Please find below a letter where we detail the modifications of our manuscript

Yours sincerely

Gentile Francesco Ficetola, Alessia Guerrieri, Isabel Cantera, Aurelie Bonin

Response to the Journal Requirements, dr. Emily Chenette, and Iain Hrynaszkiewicz

Done

The manuscript is based on data that are already accessible: the 18S-NemaBase (available at http://www.WormsEtAl.com/databases) and the GenBank database (version 249, available at https://ftp.ncbi.nlm.nih.gov/).

Furthermore, following the Editor’s suggestions, we provide the code and the data used to run analyses in figshare (10.6084/m9.figshare.24722199).

“This study was funded by the European Research Council under the European Community’s Horizon 2020 Programme, Grant Agreement no. 772284 (IceCommunities).”

“G.F.F., A.G., I.C. and A.B. have been funded by the European Research Council under the European Community’s Horizon 2020 Programme, Grant Agreement no. 772284 (IceCommunities).

Following this indication, we removed the funder information from the Acknowledgments section. The Funding Statement Section is fine.

5. PLOS requires an ORCID iD for the corresponding author in Editorial Manager on papers submitted after December 6th, 2016. Please ensure that you have an ORCID iD and that it is validated in Editorial Manager. To do this, go to ‘Update my Information’ (in the upper left-hand corner of the main menu), and click on the Fetch/Validate link next to the ORCID field. This will take you to the ORCID site and allow you to create a new iD or authenticate a pre-existing iD in Editorial Manager

Could you please change the name of the corresponding author?

The new corresponding author should be Isabel Cantera

ORCID: 0000-0003-3161-1878

isa_cantera@hotmail.com

We tried to make the link on the submission system, but the website states that Isabel Cantera is already affiliated to this manuscript. Could you please check that ever tying is ok?

We have rephrased this sentence, building on the specificity results from the in vitro analyses to strengthen our arguments and remove the reference to unpublished data.

Response to the Editor’s comments

I got the recommendations and comments from an expert reviewer in the field. Both reviewers agree that the manuscript is technically sound and the data support the conclusions.

However, the lack of an explanation in the Methods, and Results and presentation was suggested by a reviewer, and I shared the comments. Therefore, I can invite you to submit a revised version of the manuscript that addresses the points raised by the reviewers.

We modified the manuscript following the Reviewer suggestions, adding all the requested details.

Response to the Reviewer comments

Reviewer #1: The paper addresses the crucial role of nematodes as keystone actors in various ecosystems and emphasizes the limitations posed by the complexity of morphological identification for monitoring their biodiversity on a broad scale. The adoption of DNA metabarcoding has emerged as a promising approach for assessing nematode biodiversity; however, the success of this method hinges on the utilization of universal primers with high taxonomic coverage and resolution. The authors undertake a comprehensive in-silico analysis of fourteen commonly used and one newly optimized primer for nematode metabarcoding, leveraging a recently available reference database. The evaluation encompasses key parameters such as taxonomic coverage, resolution, and specificity. The findings reveal that while most primers demonstrate excellent coverage, a subset exhibits limited coverage. Notably, all primers exhibit good taxonomic resolution, particularly at higher taxonomic levels like genera or families. The study underscores that species-level resolution tends to be higher for primers amplifying longer fragments. One notable observation is the lack of high specificity for nematodes across all primers, as they amplify a substantial number of other eukaryotes. This insight emphasizes the complexity in choosing markers appropriate for nematode metabarcoding, necessitating a trade-off between taxonomic resolution and amplified fragment length. In conclusion, the in-silico analyses presented in this paper provide valuable insights into selecting optimal primers based on study goals and the origin of DNA samples. This work represents a critical step in the design and optimization of metabarcoding studies aimed at assessing nematode diversity. The introduction effectively contextualizes the importance of nematodes in ecosystems, and the paper overall contributes to advancing our understanding of nematode biodiversity using molecular techniques.

Many thanks for the positive comments

However, I think a major flaw of this study is for comparing the primer specificity. The study mainly showed the primer specificity using specificity value. But I think which base on the primer has mismatching would be more important.

Following this suggestion, we added as new analysis the weblogos, showing the mismatches between each base of the primers, and both target (nematode) and non-target taxa. These new figures have been added as figures 6-7. We expanded the methods (L 145-155), the results (L 189-191) and the discussion (L 207-211) to incorporate these new analyses

Also, I think the trade-off between taxonomic resolution and the length of amplified fragments is a key observation. The paper should provide a more detailed discussion on the practical implications of this trade-off in the context of different research goals and objectives. Recommendations for researchers navigating this trade-off would be beneficial.

Following this suggestion, we re-organized and expanded the discussion about the trade-off between taxonomic resolution and length, providing indication depending on study aim, type of substrate, and number of analyzed samples (costs-effectiveness) (L 275-286, 301-305) 

Comments on figures

As new Figure 1, It would be better to show a figure for the DNA regions in 18S with the primer regions.

Following this suggestion, we added a new figure showing the primers within the 18S rDNA region (Figure 1).

Figure 3: We acknowledge the difficulty in discerning coverage differences in Figure 3. To address this concern, we plan to enhance the visualization by adjusting the color scheme, possibly employing contrasting colors or patterns to highlight distinctions more effectively. This modification aims to provide a clearer representation of the coverage variations, ensuring that readers can easily interpret and compare the data.

Figure 4 and Others: In response to the suggestion of using different colors for the gene regions in Figure 4 and subsequent figures, we find this to be a valuable enhancement.

Following these suggestions, we changed the color palettes for Fig. 2, Fig 4 and Fig. 5. We used the discrete rainbow scheme optimized for color-blind vision (https://personal.sron.nl/~pault/#sec:qualitative). We hope differences are now clearer.

6. PLOS authors have the option to publish the peer review history of their article. If published, this will include your full peer review and any attached files.

We agree to the publication of the peer review history of the paper.

Many thanks for the useful suggestions

Francesco Ficetola, Isabel Cantera, Alessia Guerrieri, Aurelie Bonin

---

## [Editor Report · Decision Letter 1]

1 Feb 2024

In silico assessment of 18S rDNA metabarcoding markers for the characterization of nematode communities

PONE-D-23-28890R1

Dear Dr. Cantera,

We’re pleased to inform you that your manuscript has been judged scientifically suitable for publication and will be formally accepted for publication once it meets all outstanding technical requirements.

Kind regards,

Hideyuki Doi

Academic Editor

PLOS ONE

Additional Editor Comments (optional):

I carefully checked the revised manuscript as well as the response letter. I agree with the revisions and now can recommend publishing the paper.
---

## [Editor Report · Acceptance letter]

19 Mar 2024

PONE-D-23-28890R1 

PLOS ONE

Dear Dr. Cantera, 

I'm pleased to inform you that your manuscript has been deemed suitable for publication in PLOS ONE. Congratulations! Your manuscript is now being handed over to our production team.

Kind regards, 

on behalf of

Dr. Hideyuki Doi 

Academic Editor

PLOS ONE